# An observational study comparing HPV prevalence and type distribution between HPV-vaccinated and -unvaccinated girls after introduction of school-based HPV vaccination in Norway

**Espen Enerly[ID][1]\*, Ragnhild Flingtorp[1], Irene Kraus Christiansen[2], Suzanne Campbell[ID][1], Mona Hansen[2], Tor Åge Myklebust[3,4], Elisabete Weiderpass[ID][5], Mari Nygård[1]**

1 Department of Research, Cancer Registry of Norway, Oslo, Norway, 2 Department of Microbiology and Infection Control, National HPV Reference Laboratory, Akershus University Hospital, Lørenskog, Norway, 3 Department of Registration, Cancer Registry of Norway, Oslo, Norway, 4 Department of Research and Innovation, Møre and Romsdal Hospital Trust, Ålesund, Norway, 5 International Agency for Research on Cancer (IARC), World Health Organization, Lyon, France

\* espen.enerly@kreftregisteret.no

**Data Availability Statement:** Data cannot be shared publicly because the study contains many

## Abstract

### Background

Many countries have initiated school-based human papillomavirus (HPV) vaccination programs. The real-life effectiveness of HPV vaccines has become increasingly evident, especially among girls vaccinated before HPV exposure in countries with high vaccine uptake. In 2009, Norway initiated a school-based HPV vaccination program for 12-year-old girls using the quadrivalent HPV vaccine (Gardasil®), which targets HPV6, 11, 16, and 18. Here, we aim to assess type-specific vaginal and oral HPV prevalence in vaccinated compared with unvaccinated girls in the first birth cohort eligible for school-based vaccination (born in 1997).

### Methods

This observational, cross-sectional study measured the HPV prevalence ratio (PR) between vaccinated and unvaccinated girls in Norway. Facebook advertisement was used to recruit participants and disseminate information about the study. Participants self-sampled vaginal and oral specimens using an Evalyn® Brush and a FLOQSwab™, respectively. Sexual behavior was ascertained through a short questionnaire.

### Results

Among the 312 participants, 239 (76.6%) had received at least one dose of HPV vaccine prior to sexual debut. 39.1% of vaginal samples were positive for any HPV type, with similar prevalence among vaccinated and unvaccinated girls (38.5% vs 41.1%, PR: 0.93, 95% confidence interval [CI]: 0.62–1.41). For vaccine-targeted types there was some evidence of

variables with personal health information, including age, dates for vaccination, HPV test results, sexual behaviour, that would make it possible for a person to recognize her data in the dataset. In addition, the data set contained third party information from the Norwegian Immunisation Registry. The ethical committee approval, combined with the informed consent from the participants therefor does not allow us to share the data publicly. However, researchers can request access to these data by first obtaining approval for access to confidential data from Norwegian Regional Ethics Committee (https://helseforskning.etikkom.no/), and then applying to the Cancer Registry for access to the third party data used in this study (post@kreftregisteret.no).

**Funding:** The study was funded by the Cancer Registry of Norway. The funder had no role in study design, data collection and analysis, decision to publish, or preparation of the manuscript.

**Competing interests:** The authors have read the journal's policy and the authors of this manuscript have the following competing interests: M.N. received research grants from MSD/Merck through the affiliating institute.

lower prevalence in the vaccinated (0.4%) compared to the unvaccinated (6.8%) group (PR: 0.06, 95%CI: 0.01–0.52). This difference remained after adjusting for sexual behavior (PR: 0.04, 95%CI: 0.00–0.42). Only four oral samples were positive for any HPV type, and all of these participants had received at least one dose of HPV vaccine at least 1 year before oral sexual debut.

## Conclusion

There is evidence of a lower prevalence of vaccine-targeted HPV types in the vagina of vaccinated girls from the first birth cohort eligible for school-based HPV vaccination in Norway; this was not the case when considering all HPV types or types not included in the quadrivalent HPV vaccine.

## Introduction

Human papillomavirus (HPV) is a common sexually transmitted virus and infects a majority of women during their lifetime [1]. The virus is classified into high-risk and low-risk types based on oncogenic potential [2]. In the vast majority of cases, the infection is cleared without symptoms, but in some women, infection with high-risk HPV becomes persistent and causes precancerous lesions and cancer [3]. HPV16 and 18 are the cause of approximately 70% of all cervical cancers, as well as a subset of anogenital and oropharyngeal cancers [4, 5]. Three prophylactic HPV vaccines that have demonstrated high efficacy against infection with the HPV types they include are currently available on the market [6–8]. The quadrivalent vaccine (Gardasil, Merck & Co., Inc., Kenilworth, New Jersey, United States), the bivalent vaccine (Cervarix, GlaxoSmithKline Plc, Brentford, Middlesex, United Kingdom), and the nonavalent vaccine (Gardasil 9, Merck & Co., Inc., Kenilworth, New Jersey, United States). Both the quadrivalent and bivalent vaccines protect against HPV16 and 18. The quadrivalent vaccine also protects against the low-risk types HPV6 and 11, which account for approximately 90% of all genital warts [9], a condition that affects 11% of Nordic women [10].

Following the introduction of HPV vaccines, many countries initiated HPV vaccination programs [11]. The real-life effectiveness of HPV vaccines has become increasingly evident, especially in girls vaccinated before HPV exposure in countries with high vaccine uptake [12]. On a population level, the effects of HPV vaccination include a reduction in HPV prevalence, cervical cytological abnormalities, cervical histological abnormalities, and genital warts [12–14]. An observed reduction in the prevalence of vaccine-targeted HPV types in unvaccinated women also indicates a herd immunity effect [15, 16]. Moreover, although the HPV vaccine has not been approved for use for the prevention of oropharyngeal cancer, some studies have shown a reduced prevalence of vaccine-targeted HPV types in the mouth of vaccinated compared to unvaccinated girls and women [17, 18].

In 2009, Norway initiated a school-based HPV vaccination program, which vaccinated 12-year-old girls. From 2009 to autumn 2017, the quadrivalent vaccine was used; it was later replaced with the bivalent vaccine after a tender procurement process. Coverage has increased since the start of the school-based HPV vaccination program. Of those born in 2004, 83% received three vaccine doses in the 2016/2017 school year [19]. In contrast, only 65% of girls born in 1997, the first cohort to be offered the vaccine in the school-based program, had received three doses per December 2011 [20]. Catch-up vaccination was available for the period 2016–2018 for girls born in 1991 or later.

A recent study from Norway demonstrated that the prevalence of vaccine-targeted HPV types in urine was 77% (95% confidence interval [CI]: 65%-85%) lower in vaccinated compared with unvaccinated girls from the first birth cohort eligible for school-based vaccination in Norway [21]. However, at present there is no comparison of the prevalence of vaginal and oral HPV infection in vaccinated compared with unvaccinated girls in this cohort. Therefore, the overall aim of the study was to determine the effect of school-based HPV vaccination by assessing the type-specific vaginal and oral HPV prevalence in vaccinated compared with unvaccinated girls in the first birth cohort eligible for school-based HPV vaccination.

## Material and methods

### Study design

An observational, cross-sectional study design was chosen to measure the HPV prevalence ratio (PR) between vaccinated and unvaccinated girls in Norway. It is based on informed consent and was approved by the Regional Ethical Committee (#2014/2333). The study was registered nationally in HelseNorge.no [22] and in clinicaltrial.org (NCT02934724). The study included the first birth cohort eligible for the school-based HPV vaccination program, i.e., girls born in 1997 who were living in Norway in 2009, as they could provide the first sign of the effect of the vaccine on HPV prevalence.

### Recruitment of participants

The recruitment period ran from September 2016 to February 2017, when girls in the target birth cohort were 18–20 years old. This ensured that the majority of the girls were sexually active and had likely been exposed to HPV. Facebook advertisements were used to recruit participants and to disseminate information about the study. We created advertisements according to guidelines that were approved by Facebook (S1 Fig). *Pay per click* (paying when someone clicks on the ads) was chosen as the payment method. The advertisements appeared in the Facebook newsfeed of girls in the target birth cohort, and those who clicked on the advertisements were directed to a web page with an overview of the study and an online registration form on which they could provide their name, birthdate, and home address, and opt to have the study kit delivered to the nearest post office instead of their home address. Information provided by the girls in the online form was later linked to the Population Registry for validation. Only girls with validated information (matching birthdate and name, with the address used to resolve ambiguities) were sent a study kit. Invalid matches, including girls not born in 1997, received no further information. The study kit contained an informed consent form, study description, a questionnaire, a cervicovaginal sample kit, oral sample kit, and a return envelope. The sample kits contained the manufacturer's sampling instructions (translated into Norwegian for study purposes). Girls who returned the study kits along with signed informed consent were included in the study.

### Sample collection and analysis

Cervicovaginal (referred to as vaginal) samples were collected using an Evalyn® Brush, a device validated for HPV detection (Rovers Medical Devices B.V. Oss, Netherlands) [23]. At the laboratory, the Evalyn brush tips were detached from the rest of the brush using the disposable tweezers and transferred to a 12 ml tube prefilled with 4.5 ml of PreservCyt® solution (Hologic, Inc. Marlborough, Massachusetts, United States). These tubes were vortexed three times for 15 seconds, and stored at +4–7˚C overnight to let the cells detach from the brush. The suspension (400 µl) was used for DNA extraction using the automated nucleic acid

extraction platform EasyMag (EasyMag®, Biomérieux, Marcy-l'Étoile, France). Oral samples were collected using a FLOQSwab™ (#552C) (Copan Italia Spa., Brescia, Italy). The swab was transferred to 1 ml eNat buffer (Collection and Preservation System, Copan Italia Spa.) and vortexed three times for 15 seconds, 20 minutes apart, in order to let the cells detach from the swab. 400 μl were used for DNA extraction using EasyMag. HPV detection and genotyping was performed using modified general primers-PCR followed by hybridization of type-specific oligonucleotide probes (Luminex, Austin, Texas, United States), which detect and genotype 37 HPV types [24, 25].

## Human papillomavirus prevalence and sexual behavior

The objective of the study was to assess the effect of the school-based HPV vaccination program by comparing type-specific vaginal and oral HPV prevalence. Outcomes considered were prevalence of any HPV type; prevalence of individual vaccine-targeted HPV types (HPV6, 11, 16 or 18); prevalence of HPV16 or 18; and prevalence of HPV6, 11, 16, and 18 (vaccine-targeted types). The latter category was the main outcome due to the use of the quadrivalent vaccine in the school-based HPV vaccination program from 2009 to fall 2017. We also looked at the prevalence of high-risk HPV types (HPV16, 18, 31, 33, 35, 39, 45, 51, 52, 56, 58, 59, and 68) [16] and low-risk HPV types (HPV6, 11, 26, 30, 40, 42, 43, 53, 54, 61, 66, 67, 69, 70, 73, 74, 81, 82, 83, 86, 87, 89, 90, and 91), as well as non-vaccine-targeted types (positive for any of the 37 types except HPV6, 11, 16, 18).

Sexual behavior was ascertained through a short questionnaire. Two reviewers entered the responses into the study database to avoid misclassification. Using the national identification number extracted from the Population Registry, we retrieved the date and type of each dose of the complete vaccine regime from the Norwegian Immunization Registry SYSVAK. Norwegian Immunization Registry data were used, instead of self-reported vaccination status, to classify participants as vaccinated or unvaccinated (exposure classification). This was done as self-reported recollection of vaccination status may introduce recall bias [26]. To be classified as vaccinated, participants had to have received at least one dose of the quadrivalent vaccine at least 1 year before sexual debut. A small subset of our participants was vaccinated outside the school-based HPV vaccination program at an older age, i.e., after sexual debut, and among these, an even smaller subset received the bivalent instead of the quadrivalent vaccine. Girls receiving the bivalent vaccine was classified as unvaccinated. Self-reported vaccination status was used to monitor the rate of inclusion of vaccinated and unvaccinated girls and allowed us to modify recruitment towards unvaccinated girls in January 2017, as they were less sampled. Self-reported vaccination status was used for this purpose as Norwegian Immunization Registry were not retrieved at this time point in the study.

The study was not designed to estimate HPV prevalence in the general population, and therefore no specific efforts were made to recruit a nationally representative sample. However, the questionnaire allowed us in retrospect to assess if sexual behavior was similar to the general population of the same age.

## Sample size estimates

Sample size calculations for vaginal samples were based on the estimated HPV prevalence in 18-year-old Norwegian unvaccinated girls [27] and on reported HPV prevalence in vaccine efficacy trials for vaccinated girls [6]. These percentages were used to compute the number of girls needed in each group to detect the desired rate with 80% power using a 5%-level, one-sided test. This number was estimated to be approximately 100 vaccinated and 100 unvaccinated girls. Sample size calculations for oral samples were based on HPV prevalence from the

United States [28] and were computed as above. These calculations estimated that approximately 250 girls vaccinated and 250 unvaccinated girls were needed.

### Statistical methods

Chi square statistics ($\chi^2$) were used to test the association between sexual behavior and vaccination status. If one or more of the expected numbers was $\leq 5$, the Fisher's exact test was used. Multivariable Poisson regression was used to estimate PRs adjusted for lifetime number of sexual partners, age at sexual debut, and time since last sexual intercourse. Missing values were ignored. All analyses were performed in Stata 15 (StataCorp LLC, College Station, Texas, USA).

### Results

In total, 562 girls completed the online registration form and were potentially eligible for inclusion. We excluded 37 after we compared their entries with the Population Registry (Fig 1), as they reported a non-valid combination of name and birthdate (n = 13), were not born in 1997 (n = 23), or were not found in the Population Registry (n = 1). Study kits were sent to 525 potential participants; 210 were excluded because they did not return the study kit (n = 204) or the study kit contained a missing/incomplete consent form (n = 6). In total, 315 participants were confirmed eligible and were included in the study, giving a response of 56%. Among these, 312 participants were registered with a matching national identification number in the Norwegian Immunization Registry, completed the questionnaire, and were successfully tested for HPV.

Among the 312 participants, 239 (76.6%) had received at least one dose of HPV vaccine at least 1 year prior to sexual debut, and were classified as vaccinated (Table 1). The majority (95.4%) of vaccinated girls had received three doses. Vaginal and oral sexual behavior was not significantly different between vaccinated and unvaccinated girls for any of the individual questions in the questionnaire. About 13–15% of both vaccinated and unvaccinated girls were sexually naïve. The median lifetime number of sexual partners among sexually active participants was three for both vaccinated and unvaccinated girls. The majority of girls had experienced oral sexual debut (79.5% and 82.4% among vaccinated and unvaccinated girls, respectively). No significant differences between vaccinated and unvaccinated girls were observed with regard to oral sexual behavior.

The most common low-risk HPV types detected in vaginal samples were HPV90 (7.4%), HPV42 (7.1%) and HPV89 (6.1%), while HPV51 (4.2%) was the most common high-risk type

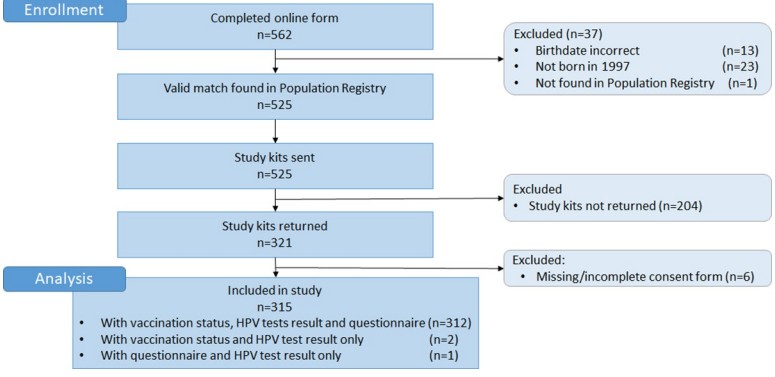

**Fig 1. CONSORT flow diagram of the study population.**

**Table 1. Baseline characteristics and questionnaire-reported sexual behavior by human papillomavirus vaccination status.**

| | Vaccinated | | Unvaccinated | | P-value | Vaccination rate | |
|---|---|---|---|---|---|---|---|
| | N | % | N | % | | % | CI |
| Total | 239 | 100 | 73 | 100 | | | |
| Vaccine doses | | | | | NA | | |
| 0 doses | 0 | 0 | 73 | 100 | | | |
| 1 dose | 1 | 0.4 | 0 | 0 | | | |
| 2 doses | 10 | 4.2 | 0 | 0 | | | |
| ≥3 doses | 228 | 95.4 | 0 | 0 | | | |
| Sexual partners | | | | | 0.604 | | |
| 0 | 31 | 13.0 | 11 | 15.1 | | 73.8 | (58.0–86.1) |
| 1 | 59 | 24.7 | 20 | 27.4 | | 74.7 | (63.6–83.8) |
| 2–5 | 86 | 36.0 | 20 | 27.4 | | 81.1 | (72.4–88.1) |
| >5 | 63 | 26.4 | 22 | 30.1 | | 74.1 | (63.5–83.0) |
| Missing | 0 | 0.0 | 0 | 0.0 | | | - |
| Age at first sexual intercourse | | 0.0 | | 0.0 | 0.342 | | |
| No debut | 31 | 13.0 | 11 | 15.1 | | 73.8 | (58.0–86.1) |
| ≤14 | 26 | 10.9 | 13 | 17.8 | | 66.7 | (49.8–80.9) |
| 15–16 | 106 | 44.4 | 26 | 35.6 | | 80.3 | (72.5–86.7) |
| ≥17 | 76 | 31.8 | 23 | 31.5 | | 76.8 | (67.2–84.7) |
| Missing | 0 | 0.0 | 0 | 0.0 | | - | - |
| Time since last sexual intercourse | | | | | 0.348 | | |
| No debut | 31 | 13.0 | 11 | 15.1 | | 73.8 | (58.0–86.1) |
| <2 days | 63 | 26.4 | 12 | 16.4 | | 84.0 | (73.7–91.4) |
| 3–7 days | 45 | 18.8 | 13 | 17.8 | | 77.6 | (64.7–87.5) |
| 1–4 weeks | 37 | 15.5 | 18 | 24.7 | | 67.3 | (53.3–79.3) |
| 5–52 weeks | 50 | 20.9 | 14 | 19.2 | | 78.1 | (66.0–87.5) |
| >1 year | 12 | 5.0 | 5 | 6.8 | | 70.6 | (44.0–89.7) |
| Missing | 1 | 0.4 | 0 | 0.0 | | - | - |
| Condom use at last sexual intercourse | | | | | 0.605 | | |
| No debut | 31 | 13.0 | 11 | 15.1 | | 73.8 | (58.0–86.1) |
| Yes | 37 | 15.5 | 11 | 15.1 | | 77.1 | (62.7–88.0) |
| No | 168 | 70.3 | 49 | 67.1 | | 77.4 | (71.3–82.8) |
| Don't know | 2 | 0.8 | 2 | 2.7 | | 50.0 | (6.8–93.2) |
| Missing | 1 | 0.4 | 0 | 0.0 | | - | - |
| Oral sexual partners | | | | | 0.109 | | |
| 0 | 42 | 17.6 | 15 | 20.5 | | 73.7 | (60.3–84.5) |
| 1 | 62 | 25.9 | 20 | 27.4 | | 75.6 | (64.9–84.4) |
| 2–5 | 99 | 41.4 | 21 | 28.8 | | 82.5 | (74.5–88.8) |
| >5 | 32 | 13.4 | 17 | 23.3 | | 65.3 | (50.4–78.3) |
| Missing | 4 | 1.7 | 0 | 0.0 | | - | - |
| Age at oral sexual debut | | | | | 0.253 | | |
| No oral sexual debut | 42 | 17.6 | 15 | 20.5 | | 73.7 | (60.3–84.5) |
| ≤14 | 17 | 7.1 | 10 | 13.7 | | 63.0 | (42.4–80.6) |
| 15–16 | 95 | 39.7 | 23 | 31.5 | | 80.5 | (72.2–87.2) |
| ≥17 | 84 | 35.1 | 25 | 34.2 | | 77.1 | (68.0–84.6) |
| Missing | 1 | 0.4 | 0 | 0.0 | | - | - |
| Time since last oral sex | | | | | 0.149 | | |
| No oral sexual debut | 42 | 17.6 | 15 | 20.5 | | 73.7 | (60.3–84.5) |

*(Continued)*

**Table 1.** (Continued)

| | Vaccinated | | Unvaccinated | | P-value | Vaccination rate | |
|---|---|---|---|---|---|---|---|
| | **N** | **%** | **N** | **%** | | **%** | **CI** |
| <2 days | 36 | 15.1 | 3 | 4.1 | | 92.3 | (79.1–98.4) |
| 3–7 days | 30 | 12.6 | 8 | 11.0 | | 78.9 | (62.7–90.4) |
| 1–4 weeks | 55 | 23.0 | 22 | 30.1 | | 71.4 | (60.0–81.2) |
| 5–52 weeks | 59 | 24.7 | 22 | 30.1 | | 72.8 | (61.8–82.1) |
| >1 year | 16 | 6.7 | 3 | 4.1 | | 84.2 | (60.4–96.7) |
| Missing | 0 | 0.0 | 0 | 0.0 | | - | - |

P-values were calculated using Chi square statistic ($\chi^2$). CI: confidence interval

(Fig 2). No significant difference between vaccinated and unvaccinated girls was detected for the individual vaccine-targeted types. Vaginal samples had a prevalence of any HPV type of 39.1% and high-risk types 19.2% with comparable prevalence among vaccinated and unvaccinated girls (38.5% vs 41.1%, PR: 0.93, 95% CI: 0.62–1.41) and (19.2% vs 19.2%, PR: 1.00, 95% CI: 0.55–1.83), respectively, (Table 2). The prevalence of the vaccine-targeted types (HPV6, 11, 16 or 18) was low overall, at 1.9% among all participants (6 positive samples). There was some evidence of lower prevalence among vaccinated (0.4%) than unvaccinated (6.8%) girls (PR: 0.06, 95% CI: 0.01–0.52), and this difference remained after adjusting for sexual behavior (PR: 0.04, 95% CI: 0.00–0.42).

Only four oral samples were positive for any HPV type (1.3%), with a similar prevalence among vaccinated and unvaccinated girls (1.3% vs 1.4%, PR: 0.92, 95% CI: 0.10–8.81). Each of these four participants had received at least one dose of HPV vaccine at least 1 year before oral sexual debut, and each was positive for a single HPV type (HPV16, 67, 69, and 87). Two were positive for the same HPV type (HPV67 and 87) in their vaginal sample, while the other two were negative for the same HPV type in their corresponding vaginal sample.

The analysis presented above for vaginal prevalence was also performed using an alternate definition of vaccinated and unvaccinated, i.e., ignoring the criteria that to be considered vaccinated, sexual debut had to occur after vaccination. This increased the number of participants included to 314 (246 vaccinated and 68 unvaccinated), as the two participants who did not

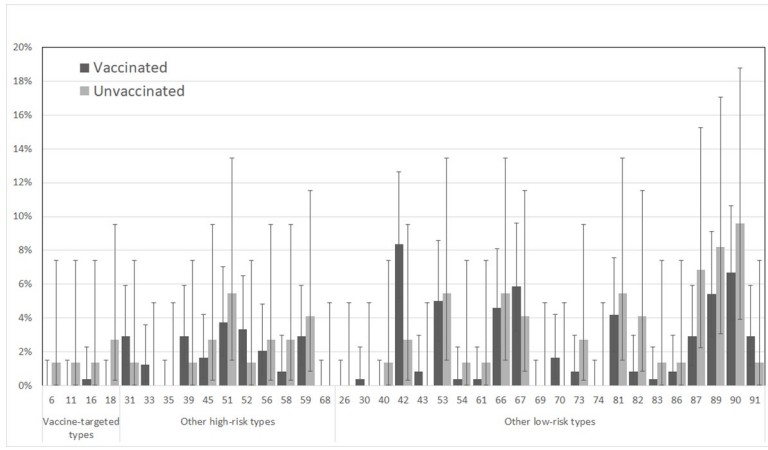

**Fig 2. Prevalence of human papillomavirus (HPV) types in vaginal samples by HPV vaccination status.** Error bars show 95% confidence intervals.

**Table 2.  Type-specific prevalence of human papillomavirus (HPV) in vaginal samples by HPV vaccination status.**

| | Prevalence (95% CI) | | | | | | Prevalence ratio (95% CI) | Adjusted prevalence ratio (95% CI) |
|---|---|---|---|---|---|---|---|---|
| | Vaccinated | | | Unvaccinated | | | | |
| | N | % | cii | N | % | cii | | |
| **Total**[*] | 239 | 100 | | 73 | 100 | | | |
| Any HPV type | 92 | 38.5 | (32.3–45.0) | 30 | 41.1 | (29.7–53.2) | 0.94 (0.62–1.41) | 0.97 (0.63–1.48) |
| HPV16 or 18 | 1 | 0.4 | (0.0–2.3) | 3 | 4.1 | (0.8–11.5) | 0.10 (0.01–0.98) | 0.11 (0.01–1.30) |
| HPV6, 11, 16 or 18 | 1 | 0.4 | (0.0–2.3) | 5 | 6.8 | (2.3–15.3) | 0.06 (0.01–0.52) | 0.04 (0.00–0.42) |
| High-risk types | 46 | 19.2 | (14.4–24.8) | 14 | 19.2 | (10.9–30.1) | 1.00 (0.55–1.83) | 1.07 (0.58–1.99) |
| Low-risk types | 77 | 32.2 | (26.3–38.5) | 23 | 31.5 | (21.1–43.4) | 1.02 (0.64–1.63) | 1.03 (0.64–1.67) |
| Non-vaccine-targeted types | 92 | 38.5 | (32.3–45.0) | 28 | 38.4 | (27.2–50.5) | 1.00 (0.66–1.53) | 1.04 (0.67–1.61) |

[*]Participants with multiple infections were counted in each category in which their type-specific HPV infection(s) belonged. CI: confidence interval

return their questionnaire (with age at sexual debut) could be included. Among the participants who shifted from unvaccinated to vaccinated, one was HPV18-positive. The difference between vaccinated and unvaccinated girls with respect to vaccine types using the original definition of vaccinated remained evident when using the alternate definition of vaccinated and unadjusted prevalence ratios (PR: 0.14, 95% CI: 0.03–0.75) (S1 Table). In addition, an *ad hoc* analysis using a definition of participants as receiving either the quadrivalent or bivalent vaccine as vaccinated had negligible impact as only one participants moved from the unvaccinated to the vaccinated category (S2 Table). A similar, *ad hoc*, analysis using a definition of vaccinated as receiving all three doses of the quadrivalent vaccine, in contrast to at least one dose, before sexual debut, gave similar trends (S3 Table).

## Discussion

Using Facebook to recruit participants born in 1997, the first birth cohort that could benefit from the Norwegian school-based HPV vaccination program, we found a lower HPV prevalence among participants who received the quadrivalent vaccine compared to unvaccinated participants. This is in line with results from clinical trials [6, 29] and observational studies in several countries [12–14]. Importantly, this is consistent with a recent study in Norway, in which approximately 18,000 urine samples from unvaccinated (1994 and 1996) and vaccinated (1997) birth cohorts were analyzed [21]. Although not directly comparable, the overall trend is similar. In the 1997 cohort, both studies showed a reduction in vaccine-targeted HPV types in vaccinated versus unvaccinated girls, while no significant differences in positivity for any HPV type or individual HPV types were observed.

We observed a lower quadrivalent HPV vaccine type prevalence among vaccinated compared to unvaccinated women. In contrast, the prevalence of any type HPV and for high-risk HPV type was comparable between vaccinated and unvaccinated. This indicates that the change is vaccine specific. As the groups performed self-sampling in the same period, the reduction is not attributable to changes in sexual behavior over time. Another interesting finding was the very low overall prevalence of vaccine-targeted HPV types in vaginal samples (1.9%, 6 out of 312 vaccinated). This is in line with a reduction in the prevalence of vaccine-targeted HPV types from urine samples from girls in birth cohorts who were not (1994 and 1996, prevalence 7.4% and 4.8%) and who were (1997, prevalence 1.4%) eligible for school-based HPV vaccination [21]. A reduction in vaccine-targeted types in unvaccinated girls was suggested to be indicative of a herd immunity effect [21] and might explain the lower type-specific prevalence we observed.

One major limitation of the present study is the small number of HPV-positive participants, which drastically reduced the power to observe differences in the prevalence of individual vaccine-targeted HPV types between vaccinated and unvaccinated participants. Our sample size estimates were based on the available HPV prevalence percentages from studies that turned out to be higher than the HPV prevalence among the participants recruited in this study [27, 28]. This might be due to an underestimation of the herd immunity effect in the sample size analysis [15].

At the end of the recruitment process, we experienced that very few girls showed interest to participate. We believe the current advertisements and recruiting strategy had reached its potential. Although, we were not aware of the overall low HPV prevalence at that time, in retrospect, the study could have benefited by including girls also born in 1998 in order to increase the sample size.

Cervicovaginal self-sampling with the Evalyn brush has previously shown good concordance with general practitioner-collected samples [30–32]; however, it might be that our participants did not sample rigorously according to the provided sampling instructions. The time- and temperature-dependent analytical stability of the Evalyn brush is high [33] and likely did not affect the overall HPV prevalence.

Positivity for any HPV type was almost absent in oral samples (1.3%), even though less than 20% of participants were oral sexually naïve. Although it is not directly comparable, oral HPV prevalence among men and women aged 16–25 years in Scotland was 2.6% (95% CI: 0.1–13.2) [34]. A study among Italian men and women aged 19–24 years showed a prevalence of 4.5% (95% CI: 2.54–7.32) [35]. A larger sample size would have been needed to increase the number of oral HPV-positive girls in our study. We chose to use the FloqSwab as we already had experience with it for cervicovaginal self-sampling in an ongoing study [36], and that preliminary data indicated in could be used for oral sampling in combination with the eNat buffers. The presence of human DNA (beta-globin) indicated that the brush was in contact with the oral mucosa, and that DNA extraction was successful, however the beta-globin level in oral samples was generally lower than that in vaginal samples. We cannot rule out that the sampling technique performed in the study affected the HPV prevalence outcome. The oral HPV prevalence should therefore be interpreted with caution, as we cannot rule out the possibility of a higher detection rate using alternative sampling methods. Our results underline the demand for a standardized oral sampling technique [37].

Targeted advertising using social networking sites, like Facebook, is rapidly increasing to recruit research participants [38, 39]. For a HPV vaccine effectiveness study in Australia, it was shown to be a rapid and cost-effective way of recruiting [40]. Inspired by the study we decided to use a similar strategy. It is particularly useful when it is desirable to raise awareness of a study in a specific demographic group and to avoid groups that are ineligible for the study [39]. The strategy of using Facebooks *Pay per click* in contrast to *pay per impression* (every time they show someone the advertisement) minimized the costs of the advertisement. In our study, Facebook recruiting allowed us to adjust the recruitment to target additional unvaccinated girls during the course of the study. Although it is a strength of the study that it recruited across the whole country, recruiting through Facebook also reduced the population that could be reached to those actively using that platform. However, at the end of 2017, 96% of girls aged 18–29 years in Norway had a Facebook account; 96% of those had an active account [41].

A strength of the study is that we could adjust for differences in sexual behavior, based on the questionnaire. However, not surprisingly, the adjusted PR was only slightly adjusted, as no significant differences between the two groups were observed. This is in line with a large Nordic study, which showed that sexual behavior is not different in vaccinated compared to unvaccinated girls [42]. The median age at both first sexual intercourse and oral sexual debut

observed in our study was 16 years. Although not directly comparable, it was fairly similar to a survey among girls aged 13–19 years from 2002, which reported 16.7 year as the median age of first sexual intercourse for girls. Although there are similarities in sexual behaviors, it should be emphasized that we did not design our study to measure representative percentages of the HPV prevalence among Norwegian girls. Another strength is that the HPV immunization history of participants was collected from the Norwegian Immunization Registry instead of relying on self-reported vaccination status. The generalizability of the main result, lowered HPV vaccine-type prevalence among vaccinated girls, is modest. The sexual behavior, vaccine coverage, and herd immunity are all factors that might differ between countries and affect the effect size.

## Conclusion

We conclude that there is evidence of a lower vaginal vaccine-targeted HPV type prevalence between vaccinated and unvaccinated girls in the first birth cohort of girls eligible for the school-based HPV vaccination program in Norway; this is not the case for the prevalence of any HPV type or non-vaccine-targeted HPV types.

## Supporting information

**S1 Fig. Facebook advertisements.** (A-B) Examples of advertisements used to target both vaccinated and unvaccinated. (C) Advertisement used to target only unvaccinated girls. Credits for images: Shutterstock.com
(TIF)

**S1 Table. Type-specific vaginal human papillomavirus (HPV) prevalence by HPV vaccination status.** A participant is defined as vaccinated if she received at least one dose of quadrivalent HPV vaccine.
(DOCX)

**S2 Table. Type-specific vaginal human papillomavirus (HPV) prevalence by HPV vaccination status.** A participant is defined as vaccinated if she received at least one dose of the bivalent or the quadrivalent HPV vaccine at least the calendar year before sexual debut.
(DOCX)

**S3 Table. Type-specific vaginal human papillomavirus (HPV) prevalence by HPV vaccination status.** A participant is defined as vaccinated if she received at least three doses the quadrivalent HPV vaccine at least the calendar year before sexual debut.
(DOCX)

**S4 Table. STROBE checklist for cross sectional studies.** STROBE Statement—Checklist of items that should be included in reports of cross-sectional studies.
(DOCX)

## Acknowledgments

The authors would like to thank the girls who participated in the study. The authors would like to thank Trudy Perdrix-Thoma for editorial assistance. The authors would also like to acknowledge the Copan Italia Spa., Brescia, Italy for providing the FLOQSwab™ and the majority of the eNat buffer free of charge.

## Author Contributions

**Conceptualization:** Espen Enerly, Elisabete Weiderpass, Mari Nygård.

**Data curation:** Ragnhild Flingtorp, Suzanne Campbell.

**Formal analysis:** Irene Kraus Christiansen, Suzanne Campbell, Mona Hansen, Tor Åge Myklebust.

**Funding acquisition:** Mari Nygård.

**Investigation:** Espen Enerly, Irene Kraus Christiansen, Mona Hansen.

**Methodology:** Tor Åge Myklebust.

**Project administration:** Espen Enerly, Ragnhild Flingtorp, Suzanne Campbell.

**Supervision:** Elisabete Weiderpass, Mari Nygård.

**Writing – original draft:** Espen Enerly.

**Writing – review & editing:** Espen Enerly, Ragnhild Flingtorp, Irene Kraus Christiansen, Suzanne Campbell, Mona Hansen, Tor Åge Myklebust, Elisabete Weiderpass, Mari Nygård.

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
