## [Decision Letter · Decision Letter 0]

2 Jul 2019

PONE-D-19-15806

An observational study comparing HPV prevalence and type distribution between HPV-vaccinated and -unvaccinated girls after introduction of school-based HPV vaccination in Norway

PLOS ONE

Dear Dr Enerly,

Thank you for submitting your manuscript to PLOS ONE. After careful consideration, we feel that it has merit but does not fully meet PLOS ONE’s publication criteria as it currently stands. Therefore, we invite you to submit a revised version of the manuscript that addresses the points raised during the review process.

We would appreciate receiving your revised manuscript by Aug 16 2019 11:59PM. To enhance the reproducibility of your results, we recommend that if applicable you deposit your laboratory protocols in protocols.io, where a protocol can be assigned its own identifier (DOI) such that it can be cited independently in the future. For instructions see: http://journals.plos.org/plosone/s/submission-guidelines#loc-laboratory-protocols

We look forward to receiving your revised manuscript.

Kind regards,

Maria Lina Tornesello

Academic Editor

PLOS ONE

**Journal Requirements:**

2. We ask that you please include your ethics statement and IRB approval information in the methods section of your manuscript.

" We have read the journal's policy and the authors of this manuscript have the following competing interests: M.N. has received research grants from MSD/Merck through the affiliating institute.

The other authors have declared that no competing interests exist.

WHO-IARC: Where authors are identified as personnel of the International Agency for Research on Cancer / World Health Organization, the authors alone are responsible for the views expressed in this article and they do not necessarily represent the decisions, policy or views of the International Agency for Research on Cancer / World Health Organization.

Norwegian Immunization Registry SYSVAK (Norwegian Institute of Public Health) provided data, but is not responsible for the analyses and interpretation in this article."

**Comments to the Author**

1. Is the manuscript technically sound, and do the data support the conclusions?

Reviewer #1: Partly

Reviewer #2: No

Reviewer #3: Yes

2. Has the statistical analysis been performed appropriately and rigorously? 

Reviewer #1: Yes

Reviewer #2: Yes

Reviewer #3: Yes

3. Have the authors made all data underlying the findings in their manuscript fully available?

Reviewer #1: Yes

Reviewer #2: Yes

Reviewer #3: Yes

4. Is the manuscript presented in an intelligible fashion and written in standard English?

Reviewer #1: Yes

Reviewer #2: Yes

Reviewer #3: Yes

5. Review Comments to the Author

Reviewer #1: This is an interesting study, which confirms what has previously been seen in Norway with respect to impact of the HPV vaccine programme. Unfortunately, due to the small numbers of the females involved in the study, it is very difficult to see what more such a paper adds to the previously referenced studies from Norway and elsewhere. The utility of using Facebook to recruit women into providing samples and sexual behaviour data is of interest and it is that methodology that I think the authors could focus on more, as a potential for surveillance expansion. However, in its current format, I don't believe the data are particularly novel or representative of a population. The following may help to revise the manuscript.

Questions/comments

1. What has been the uptake of the HPV vaccine in Norway since the programme started? This is not included.

2. Why did Norway move from 4v to 2v? This is not explained and at odds with most countries e.g. UK - the reverse occurred.

3. Please expand on the Facebook recruitment strategy targeting the specific birth cohorts - this is interesting and will be of interest to other countries. Especially if samples are then provided via urine, which has been shown to have similar data with respect to HPV detection, when compared to clinically taken smears.

4. Why are girls vaccinated with bivalent described as unvaccinated? This is not appropriate.

5. Why do the authors change their vaccine status identification from registry to self-reported in the Methods? This is incongruous.

6. Be careful using the term 'rate' with respect to vaccine uptake - they are only rates if a time component is included - otherwise percentage uptake is sufficient and preferable.

7. The authors state that HPV 89 and 90 were most prevalent low-risk types - but no percentages included. Please include.

8. I think it is important to state that while 'vaccine-specific' HPV has reduced, other HR-HPV has not - this suggests a vaccine effect rather than any change in sexual behaviour over time. The authors should develop this.

Reviewer #2: Review Enerly et al.

An observational study comparing HPV prevalence and type distribution between HPV-vaccinated and – unvaccinated girls after a school-based HPV vaccination in Norway

The authors performed a cross sectional study of the HPV prevalence comparing vaccinated and non vaccinated girls in Norway. They used an Evalyn Brush and FLOQSwab device to retrieve material cellular material from the participants from either cervix and oral mucosa. They found a lower prevalence of HPV infections in the vaccinated girls compared to the unvaccinated. The could only ensure at least one vaccination of the participants, which was retrieved from the national epidemiology files.

Major concerns:

The authors present an approved study from the ethics commission and registered the study at the NCT files. However it is doubtful if the participants recruited via facebook are a representative group in this age group. In addition, one must doubt if all participants are able to handle the Evalyn device appropriate. How can the authors ensure that they do observe either false positive or false negative data. To my opinion, controlled studies that are published should ensure that participants see a physician who can ensure that the participant identify herself as the person that take part in the study and that the specimen is taken by an experienced physician under controlled circumstances.

Furthermore, it is to my opinion critical that the authors can only ensure that the girls have received at least only one vaccine shot. Why isn’t it possible that they have all information about the full vaccination program. This is a major setback in the design of the study.

Not surprising the authors have to exclude 210 girls because they did not send back the kit the consent was incomplete. This underscores the major disadvantage of the presented approach the authors have chosen.

Reviewer #3: The sample size is small. But the authors have no commented on this. It is suggested to include justification for the small sample size, in the text. May be it can be included as limitation of the study.

6. PLOS authors have the option to publish the peer review history of their article (what does this mean?). If published, this will include your full peer review and any attached files.

Reviewer #1: Yes: Kevin Pollock

Reviewer #2: No

Reviewer #3: No

---

## [Author Response · Author response to Decision Letter 0]

19 Aug 2019

Response to reviewers

Reviewer #1

Reviewer #1: This is an interesting study, which confirms what has previously been seen in Norway with respect to impact of the HPV vaccine programme. Unfortunately, due to the small numbers of the females involved in the study, it is very difficult to see what more such a paper adds to the previously referenced studies from Norway and elsewhere. The utility of using Facebook to recruit women into providing samples and sexual behaviour data is of interest and it is that methodology that I think the authors could focus on more, as a potential for surveillance expansion. However, in its current format, I don't believe the data are particularly novel or representative of a population. The following may help to revise the manuscript.

1. What has been the uptake of the HPV vaccine in Norway since the programme started? This is not included.

Answer: We have added this information to the Introduction along with two references.

2. Why did Norway move from 4v to 2v? This is not explained and at odds with most countries e.g. UK - the reverse occurred. 

Answer: The change was done after a tender for procurement (In Norwegian at https://www.doffin.no/en/notice/details/2016-175572). The evaluation of the offers is not publicly available, but the evaluation weighted “Price 60% and Quality (Effect and safety) 40% (Part IV 2.1 in the link above). We have added a note to the Introduction about the tender.

3. Please expand on the Facebook recruitment strategy targeting the specific birth cohorts - this is interesting and will be of interest to other countries. Especially if samples are then provided via urine, which has been shown to have similar data with respect to HPV detection, when compared to clinically taken smears.

Answer: We have expanded on the use of Facebook both in Recruitment of participants chapter and in the discussion. In addition supplementary Figure 1 is updated with more examples on the advertsiments

4. Why are girls vaccinated with bivalent described as unvaccinated? This is not appropriate.

Answer: We certainly agree that treating bivalent vaccinated girls as unvaccinated is not appropriate per se. However, we did not foresee that Norway would change to the bivalent vaccine. When creating the analysis plan we choose the definition of vaccinated as “Women born in 1997, resident in Norway in 2009, vaccinated with the quadrivalent HPV vaccine”. This was used to describe the definition in the study in ClinicalTrial.org (https://clinicaltrials.gov/ct2/show/NCT02934724?term=enerly&rank=1). Therefore we believe the data should be analysed using the definition defined upfront of the study. However, we have now included an ad hoc analysis. Adding bivalent vaccination to the definition moves only one participant from the unvaccinated category to the vaccinated category. This has negligible impact of the results. We have added this information to the results as a supplementary table.

5. Why do the authors change their vaccine status identification from registry to self-reported in the Methods? This is incongruous.

Answer: We used self-reported vaccination status only to optimize the recruitment process. We noticed that few participants reported to be vaccinated and therefore selectively recruited unvaccinated for a short period. During recruitment we did not have access to registry (sysvak) information. To clarify this use we have added a sentence in the method chapter.

6. Be careful using the term 'rate' with respect to vaccine uptake - they are only rates if a time component is included - otherwise percentage uptake is sufficient and preferable.

Answer: We have updated to percentage throughout the paper when appropriate. 

7. The authors state that HPV 89 and 90 were most prevalent low-risk types - but no percentages included. Please include.

Answer: Percentages are now included. HPV42 was missing and is now included too.

8. I think it is important to state that while 'vaccine-specific' HPV has reduced, other HR-HPV has not - this suggests a vaccine effect rather than any change in sexual behaviour over time. The authors should develop this.

Answer: We have added a sentence to the result to highlight this and we discuss it more thoroughly in the discussion. 

Reviewer #2: Review Enerly et al.

An observational study comparing HPV prevalence and type distribution between HPV-vaccinated and – unvaccinated girls after a school-based HPV vaccination in Norway

The authors performed a cross sectional study of the HPV prevalence comparing vaccinated and non vaccinated girls in Norway. They used an Evalyn Brush and FLOQSwab device to retrieve material cellular material from the participants from either cervix and oral mucosa. They found a lower prevalence of HPV infections in the vaccinated girls compared to the unvaccinated. The could only ensure at least one vaccination of the participants, which was retrieved from the national epidemiology files.

Answer:

Major concerns:

The authors present an approved study from the ethics commission and registered the study at the NCT files. However it is doubtful if the participants recruited via facebook are a representative group in this age group. 

Answer: Although some similar sexual behaviour is seen comparing data from the questionnaire with data from other population-based studies of Norwegian women, we agree that there is likely a bias when recruiting through Facebook. We have therefore commented on this issue which we also touch upon in the last sentences of the method chapter “Human papillomavirus prevalence and sexual behaviour”. 

In addition, one must doubt if all participants are able to handle the Evalyn device appropriate. How can the authors ensure that they do observe either false positive or false negative data. To my opinion, controlled studies that are published should ensure that participants see a physician who can ensure that the participant identify herself as the person that take part in the study and that the specimen is taken by an experienced physician under controlled circumstances.

Answer: We chose not to design the study to require the girls to see an experienced physician under controlled circumstances. Firstly, because the recruitment process would likely result in fewer participants and higher costs. Secondly, both vaccinated and unvaccinated were sampled the same way and reduced efficiency or contamination would likely be evenly distributed between the groups. However, we agree that we cannot ensure that girls sample correctly according to the instruction. Still, taken together we believe the study design is valid for answering the aims of the study.

Furthermore, it is to my opinion critical that the authors can only ensure that the girls have received at least only one vaccine shot. Why isn’t it possible that they have all information about the full vaccination program. This is a major setback in the design of the study.

Answer: Thanks for pointing it out that it was unclear as we in fact have the information. In the chapter “Human papillomavirus prevalence and sexual behavior” we have modified the sentence to clarify this. The distribution of the number of doses is listed in Table 1. An Ad hoc analysis using the definition of vaccinated as receiving all three doses at least the year before sexual debut was done and is added to the results and as a supplementary table.

Not surprising the authors have to exclude 210 girls because they did not send back the kit the consent was incomplete. This underscores the major disadvantage of the presented approach the authors have chosen.

Answer: Ideally, we wish that all girls that showed interest for the study would consent to the study after receiving the kits. However, that 60% consented and returned the kits we believe is still reasonable. The aim is not to measure the general HPV prevalence. Therefore we believe the return percentage is acceptable. The extra cost is of course a drawback. The two groups that finally consented to the study did show similar sexual behavior patterns and we believe therefore the 60% returns did not introduce biases between the groups that we could not adjust for in the analysis.

Reviewer #3: The sample size is small. But the authors have no commented on this. It is suggested to include justification for the small sample size, in the text. May be it can be included as limitation of the study.

Answer: We have expanded on this limitation in the discussion.

---

## [Editor Report · Decision Letter 1]

25 Sep 2019

An observational study comparing HPV prevalence and type distribution between HPV-vaccinated and -unvaccinated girls after introduction of school-based HPV vaccination in Norway

PONE-D-19-15806R1

Dear Dr. Enerly,

We are pleased to inform you that your manuscript has been judged scientifically suitable for publication and will be formally accepted for publication once it complies with all outstanding technical requirements.

With kind regards,

Maria Lina Tornesello

Academic Editor

PLOS ONE
---

## [Editor Report · Acceptance letter]

2 Oct 2019

PONE-D-19-15806R1 

An observational study comparing HPV prevalence and type distribution between HPV-vaccinated and -unvaccinated girls after introduction of school-based HPV vaccination in Norway 

Dear Dr. Enerly:

I am pleased to inform you that your manuscript has been deemed suitable for publication in PLOS ONE. Congratulations! Your manuscript is now with our production department. 

With kind regards,

on behalf of

Dr. Maria Lina Tornesello 

Academic Editor

PLOS ONE